# Learning Disconnected Manifolds: Avoiding The No GAN's Land By Latent Rejection

## Abstract

Standard formulations of GANs, where a continuous function deforms a connected latent space, have been shown to be misspecified when fitting disconnected manifolds. In particular, when covering different classes of images, the generator will necessarily sample some low quality images in between the modes. Rather than modify the learning procedure, a line of works aims at improving the sampling quality from trained generators. Thus, it is now common to introduce a rejection step within the generation procedure. Building on this, we propose to train an additional network and transform the latent space via an adversarial learning of importance weights. This idea has several advantages: 1) it provides a way to inject disconnectedness on any GAN architecture, 2) since the rejection happens in the latent space, it avoids going through both the generator and the discriminator saving computation time, 3) this *importance weights* formulation provides a principled way to reduce the Wasserstein's distance to the target distribution. We demonstrate the effectiveness of our method on different datasets, both synthetic and high dimensional.

## 1 Introduction

GANs (Goodfellow et al., 2014) are an effective way to learn complex and high-dimensional distributions, leading to state-of-the-art models for image synthesis in both unconditional (Karras et al., 2019) and conditional settings (Brock et al., 2019). However, it is well-known that a single generator with a unimodal latent variable cannot recover a distribution composed of disconnected sub-manifolds (Khayatkhoei et al., 2018). This leads to a common problem for practitioners: the necessary existence of very-low quality samples when covering different modes. This is formalized by Tanielian et al. (2020) which refers to this area as the no GAN's land and provides impossibility theorems on the learning of disconnected manifolds with standard formulations of GANs. Fitting a disconnected target distribution requires an additional mechanism inserting disconnectedness in the modeled distribution. A first solution is to add some expressivity to the model: Khayatkhoei et al. (2018) propose to train a mixture of generators while Gurumurthy et al. (2017) make use of a multi-modal latent distribution. A second solution is to improve the quality of a trained generative model by avoiding its poorest samples (Tao et al., 2018; Azadi et al., 2019; Turner et al., 2019; Grover et al., 2019; Tanaka, 2019).

This second line of research relies heavily on a variety of Monte-Carlo algorithms, such as Rejection Sampling or the Metropolis-Hastings. These methods aim at sampling from a target distribution, while having only access to samples generated from a proposal distribution. This idea was successfully applied to GANs, using the previously learned generative distribution as a proposal distribution. However, one of the main drawback is that Monte-Carlo algorithms only guarantee to sample from the target distribution under strong assumptions. First, we need access to the density ratios between the proposal and target distributions or equivalently to a perfect discriminator (Azadi et al., 2019). Second, the support of the proposal distribution must fully cover the one of the target distribution, which means no mode collapse. This is known to be very demanding in high dimension since the intersection of supports between the proposal and target distribution is likely to be negligible (Arjovsky and Bottou, 2017, Lemma 3). In this setting, an optimal discriminator would give null acceptance probabilities for almost any generated points, leading to a lower performance.

To tackle the aforementioned issue, we propose a novel method aiming at reducing the Wasserstein distance between the previously trained generative model and the target distribution. This is done via

the adversarial training of a third network that learns importance weights in the latent space. The goal is to learn the redistribution of mass of the modeled distribution that best fits the target distribution. To better understand our approach, we first consider a simple 2D motivational example where the real data lies on four disconnected manifolds. To approximate this, the generator splits the latent space into four distinct areas and maps data points located in the frontiers, areas in orange in Figure 1b, out of the true manifold (see Figure 1a). Our method consequently aims at learning latent importance weights that can identify these frontiers and simply avoid them. This is highlighted in Figure 1d where the importance weighter has identified these four frontiers. When sampling from the new latent distribution, we can now perfectly fit the mixture of four gaussians (see Figure 1c).

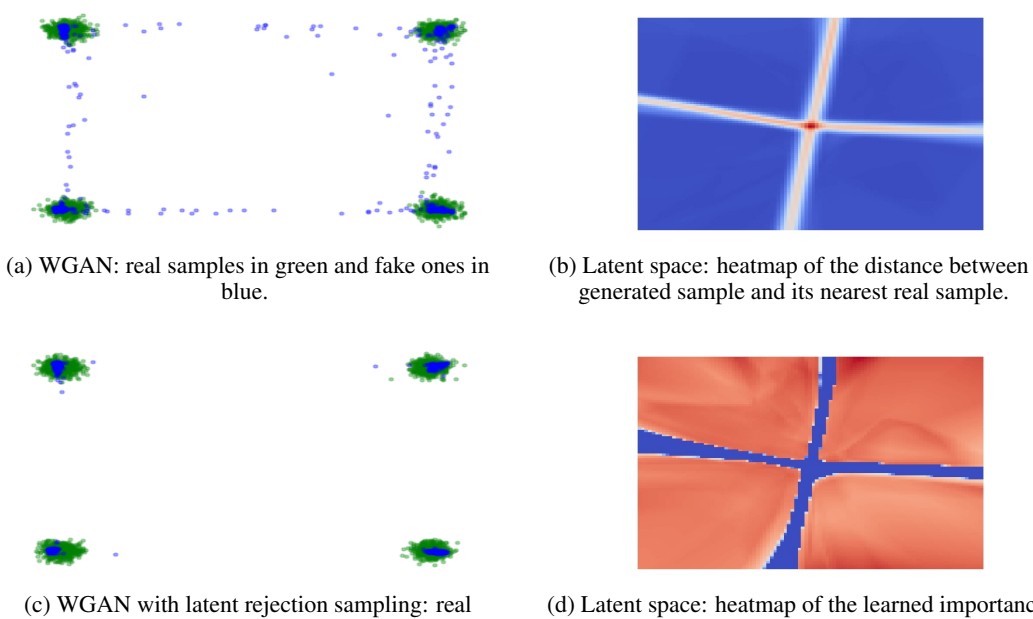

(a) WGAN: real samples in green and fake ones in blue.

(b) Latent space: heatmap of the distance between a generated sample and its nearest real sample.

(c) WGAN with latent rejection sampling: real samples in green and fake ones in blue.

(d) Latent space: heatmap of the learned importance weights. The blue frontiers have zero weights.

Figure 1: Learning disconnected manifolds leads to the apparition of an area in the latent space generating points outside the target manifold. With the use of the importance weighter, one can avoid this specific area and better fit the target distribution.

Our contributions are the following:

- We discuss works improving the sampling quality of GANs and identify their limitations.
- We propose a novel approach that directly modifies the latent space distribution. It provides a principled way to reduce the Wasserstein distance to the target distribution.
- We thoroughly compare our method with a large set of previous approaches on a variety of datasets and distributions. We empirically show that our solution significantly reduces the computational cost of inference while demonstrating an equal efficiency.

**Notation.** Before moving to the related work section, we shortly present notation needed in the paper. The goal of the generator is to generate data points that are "similar" to samples collected from some target probability measure $\mu_\star$. The measure $\mu_\star$ is defined on a potentially high dimensional space $\mathbb{R}^D$, equipped with the euclidean norm $\|\cdot\|$. To approach $\mu_\star$, we use a parametric family of generative distribution where each distribution is the push-forward measure of a latent distribution $Z$ and a continuous function modeled by a neural network. In most of all practical applications, the random variable $Z$ defined on a low dimensional space $\mathbb{R}^d$ is either a multivariate Gaussian distribution or uniform distribution. The generator is a parameterized class of functions from $\mathbb{R}^d$ to $\mathbb{R}^D$, say $\mathscr{G} = \{G_\theta : \theta \in \Theta\}$, where $\Theta \subseteq \mathbb{R}^p$ is the set of parameters describing the model. Each function $G_\theta$ takes input from $Z$ and outputs "fake" observations with distribution $\mu_\theta = G_\theta \sharp Z$. On the other hand, the discriminator is described by a family of functions from $\mathbb{R}^D$ to $\mathbb{R}$, say $\mathscr{D} = \{D_\alpha : \alpha \in \Lambda\}$, $\Lambda \subseteq \mathbb{R}^Q$, where each $D_\alpha$. Finally, for any given distribution $\mu$, we note $S_\mu$ its support.

## 2 RELATED WORK

### 2.1 DISCONNECTED MANIFOLD LEARNING: HOW TO TRAIN AND EVALUATE GANS

Goodfellow et al. (2014) already stated that when training vanilla GANs, the generator could ignore modes of the target distribution: this is the mode collapse. A significant step towards understanding this phenomenon was made by Arjovsky and Bottou (2017) who explained that the standard formulation of GANs leads to vanishing or unstable gradients. The authors proposed the Wasserstein GANs (WGANs) architecture (Arjovsky et al., 2017) where, in particular, discriminative functions are restricted to the class of 1-Lipschitz functions. WGANs aim at solving:

$$\sup_{\alpha \in A} \inf_{\theta \in \Theta} \mathbb{E}_{x \sim \mu_\star} D_\alpha(x) - \mathbb{E}_{z \sim Z} D_\alpha(G_\theta(z))) \tag{1}$$

The broader drawback of standard GANs is that, since any modeled distribution is the push-forward of a unimodal distribution by a continuous transformation, it consequently has a connected support. This means that when the generator covers multiple disconnected modes of the target distribution, it necessarily generates samples out of the real data manifold (Khayatkhoei et al., 2018). Consequently, any thorough evaluation of GANs should assess simultaneously both the quality and the variety of the generated samples. Sajjadi et al. (2018) argue that a single-digit metric such as the Inception Score (Salimans et al., 2016) or the Frechet Inception distance (Heusel et al., 2017) is thus not adequate to compare generative models. To solve this issue, the authors propose a Precision/Recall metric that aims at measuring both the *mode dropping* and the *mode inventing*.

In the Improved Precision/Recall (Kynkäänniemi et al., 2019), the precision refers to the portion of generated points that belongs to the target manifold, while the recall measures how much of the target distribution can be re-constructed by the model distribution. Building on this metric, Tanielian et al. (2020) highlighted the trade-off property of GANs deriving upper-bounds on the precision of standard GANs. To solve this problem, a common direction of research consists in over-parameterizing the generative model. Khayatkhoei et al. (2018) enforce diversity by using a mixture of generators while Gurumurthy et al. (2017) suggest that a mixture of Gaussians in the latent space is efficient to learn diverse and limited data.

### 2.2 IMPROVING THE QUALITY OF TRAINED GENERATORS

To better fit disconnected manifolds with standard GANs architectures, another line of research consists in inserting disconnectedness into a previously learned generative distribution $\mu_\theta$. Tanielian et al. (2020) proposed an heuristic to remove the no GAN's land (*i.e.* samples mapped out of the true manifold): rejecting data points with a high Jacobian Frobenius norm. Another possibility would be to use one of the different Monte-Carlo methods (Robert and Casella, 2013) and apply it to GANs. Building up on the well-known inference theory, Azadi et al. (2019) suggests the use of rejection sampling to improve the quality of the proposal distribution $\mu_\theta$. One can compute density ratios using either a classifier trained from scratch or the discriminator obtained at the end of the training. Consequently, in this Discriminator Rejection Sampling (DRS), any generated data point $x \sim \mu_\theta$ is accepted with the following acceptance probability $\mathbb{P}_a$:

$$\mathbb{P}_a(x) = \frac{\mu_\star(x)}{M\mu_\theta(x)} \quad \text{where} \quad M = \max_{x \in S_{\mu_\theta}} \frac{\mu_\star(x)}{\mu_\theta(x)}, \tag{2}$$

where $\mu_\star$ and $\mu_\theta$ here refers to the density functions. Similarly, Turner et al. (2019) use the same density ratios and derive MH-GAN, an adaptation of the Metropolis-Hasting algorithm (Hastings, 1970), that improves the sampling from $\mu_\theta$. Finally, Grover et al. (2019) use these density ratios $r$ as importance weights and define an importance resampled generative model whose density is now defined by $\hat{\mu}_\theta \propto \mu_\theta \times r(x)$. In order to perform discrete sampling from $\hat{\mu}_\theta$, authors rely on the Sampling-Importance-Resampling (SIR) algorithm (Rubin, 1988; Liu and Chen, 1998). This defines a new distribution $\hat{\mu}_\theta^{\text{SIR}}$:

$$\hat{\mu}_\theta^{\text{SIR}}(x_i) = \frac{r(x_i)}{\sum\limits_{j=1}^{n} r(x_j)} \quad \text{where} \quad x_1, \ldots, x_n \sim \mu_\theta^n.$$

Note that these algorithms rely on the same density ratios and an acceptance-rejection scheme. In Rejection Sampling, the acceptance rate is uncontrollable but sampling from $\mu_\star$ is assured. With SIR and MH, the acceptance rate is controllable but sampling from $\mu_\star$ is no longer guaranteed.

## 3 ADVERSARIAL LEARNING OF LATENT IMPORTANCE WEIGHTS

### 3.1 OUR APPROACH

Similar to previous works, our method consists in improving the performance of a given generative model, post-training. Given a trained WGANs $(G_\theta, D_\alpha)$, we now propose to learn importance weights in the latent space. To do so, we use a feed-forward neural network from $\mathbb{R}^d$ to $\mathbb{R}^+$, say $\Omega = \{w_\varphi : \varphi \in \Phi\}$. The neural network $w_\varphi$ is trained using an adversarial process with the discriminator $D_\alpha$, whilst keeping the weights of $G_\theta$ frozen. We now want to solve the following:

$$\sup_{\alpha \in A} \inf_{\varphi \in \Phi} \mathbb{E}_{x \sim \mu_\star} D_\alpha(x) - \mathbb{E}_{z \sim Z}\big(w_\varphi(z) \times D_\alpha(G_\theta(z)))\big) \tag{3}$$

Note that our formulation can also be plugged on top of many different objective functions. Interestingly, the use of the predictor $w_\varphi$ defines a new latent space distribution whose density $\hat{\gamma}$ is defined by $\hat{\gamma}(z) \propto w_\varphi(z) \times \gamma(z)$. Consequently, the newly defined modeled distribution $\hat{\mu}_\theta$ is defined as the pushforward $\hat{\mu}_\theta = G_\theta \sharp \hat{\gamma}$. The proposed method can be seen as minimizing the Wasserstein distance to the target distribution, over an increased class of generative distributions. The network $w_\varphi$ thus learns how to redistribute the mass of $\mu_\theta$ such that $\hat{\mu}_\theta$ is closer to $\mu_\star$ in terms Wasserstein distance.

However, as in the field of counterfactual estimation, a naive optimization of importance weights by gradient descent can lead to trivial solutions. First, if for example, the Wasserstein critic $D_\alpha$ outputs negative values for any generated samples, the network $w_\varphi$ could simply learn to avoid the dataset and output 0 everywhere. To avoid this issue, we follow Swaminathan and Joachims (2015c) and scale the output of the discriminator such that the *reward* is always positive. A second problem comes from the fact that equation 3 can now be minimized not only by putting large importance weights $w_\varphi(z)$ on the examples with high likelihoods $D_\alpha(G(z))$, but also by maximizing the sum of the weights: this is the propensity overfitting (Swaminathan and Joachims, 2015a). To stabilize the optimisation process, we consequently introduce two important regularization techniques:

**Self-normalization.** Similarly to Swaminathan and Joachims (2015a), we advocate the use of a normalization of the importance weights. To be more precise, we enforce the expectation of the importance weights to be close 1 by adding a penalty term. By doing so, we prohibit the propensity overfitting since the sum of the importance weights in the batch is bounded.

**Soft-Clipping** To avoid cases where small areas of $z$ have really high $w_\varphi(z)$ values, which would lead to mode collapse, we enforce a soft-clipping on the weights (Bottou et al., 2013; Grover et al., 2019). Note that this constraint on $w_\varphi(z)$ could also be implemented with a bounded activation function on the final layer, such as a re-scaled sigmoid or tanh activation.

Finally, we thus get the following objective function:

$$\sup_{\varphi \in \Phi} \mathbb{E}_{z \sim Z} \underbrace{w_\varphi(z)\big(D_\alpha(G_\theta(z))\big) - \nabla}_{\text{discriminator reward}} - \lambda_1 \underbrace{\big(\mathbb{E}_{z \sim Z} w_\varphi(z) - 1\big)^2}_{\text{self-normalization}} - \lambda_2 \underbrace{\mathbb{E}_{z \sim Z} \max\big(0, (w_\varphi(z) - m)\big)^2}_{\text{soft-clipping}}, \tag{4}$$

where $\nabla = \min_{z \sim Z} D_\alpha(G(z))$. $\lambda_1$, $\lambda_2$, and $m$ are hyper-parameters (values displayed in Appendix).

### 3.2 SAMPLING FROM THE NEW DISTRIBUTION

As mentionned above, the scale and variance of the learned importance weights are actively controlled, as it is done in counterfactual estimation (Bottou et al., 2013; Swaminathan and Joachims, 2015b; Faury et al., 2020). Doing so, we explicitly control the acceptance rates $\mathbb{P}_a(z)$ of the rejection sampling algorithm performed on $\hat{\gamma}$, since for any given $z \sim Z$, we have:

$$\mathbb{P}_a(z) = \frac{\hat{\gamma}(z)}{m\gamma(z)} \quad \text{and} \quad \mathbb{E}_Z \, \mathbb{P}_a(z) = \int_{\mathbb{R}^d} \frac{\hat{\gamma}(z)}{m\gamma(z)} \gamma(z) dz = \int_{\mathbb{R}^d} \frac{\hat{\gamma}(z)}{m} dz = \frac{1}{m},$$

where $m$ is the maximum output of the importance weighter as defined in equation 4. We define as Latent Rejection Sampling (latentRS), the method that performs the Rejection Sampling algorithm on top of the learned importance weights. Since the exact sampling of the distribution $\hat{\gamma}$ is now tractable with a rejection sampling algorithm, we do not need to implement neither the Metropolis-Hasting nor the Sampling Importance Resampling algorithm.

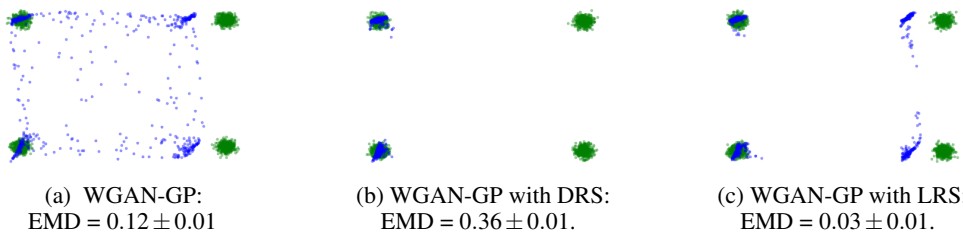

(a) WGAN-GP:
EMD = $0.12 \pm 0.01$

(b) WGAN-GP with DRS:
EMD = $0.36 \pm 0.01$.

(c) WGAN-GP with LRS:
EMD = $0.03 \pm 0.01$.

Figure 2: In this synthetic experiment, the target distribution has two modes slightly shifted from the generated distribution. When using an optimal discriminator, DRS only selects the intersection of the two supports, thus removing important data. Our method achieves a much better fit and does not suppress any mode. The EMD metric confirms the interest of the method in this specific case.

Inspired from the literature on latent space optimal transport (Salimans et al., 2018; Agustsson et al., 2019; Tanaka, 2019), we also propose a second method where we perform gradient ascent in the latent space. To be more precise, for any given sample in the latent space, we follow the path maximizing the learned importance weights. This method is denoted latent Gradient Ascent (latentGA). In high-dimension, similarly to Tanaka (2019, Algorithm 2), gradients are projected to restrict $z$ on the training support. Note that the learning rate and the number of updates used for this method are hyper-parameter that need to be tuned.

### 3.3 ADVANTAGES OF THE PROPOSED APPROACH

We now discuss, in detail, the flaws of previous Monte-Carlo based approaches:

**1) Computational cost.** By using sampling algorithms in the latent space, we avoid going through both the generator and the discriminator, leading to a significant computational speed-up. This is of particular interest when dealing with high-dimensional spaces since we do not need to pass through deep CNNs generator and discriminator (Brock et al., 2019).

**2) Existence of density functions.** Every Monte-Carlo based methods assume that both $\mu_\star$ and $\mu_\theta$ are probability distributions with associated density functions. However, in high dimension, the hypothesis that data tend to lie near a low dimensional manifold (Fefferman et al., 2016) is now commonly accepted. Besides, it is often the case that GANs be defined as the push-forward from much lower dimensional space, that is $d << D$. In that case, neither $\mu_\star$ nor $\mu_\theta$ have density function in $\mathbb{R}^D$. Note that our method based on Wasserstein distance does not require this assumption.

**3) Covering of the support $S_{\mu_\star}$.** First, Monte-Carlo methods are well-known to suffer from the curse of dimensionality (Mengersen et al., 1996; Robert and Casella, 2013). Besides, in the context of GANs, Arjovsky and Bottou (2017, Theorem 2.2) have shown that the intersection $S_{\mu_\star} \bigcap S_{\mu_\theta}$ is likely to be a negligible set under $\mu_\theta$. In this specific case, the density ratios would evaluate close to 0 almost everywhere on $S_{\mu_\theta}$, increasing the time complexity. More generally, Monte-Carlo based methods tend to avoid any area within $S_{\mu_\theta} \backslash S_{\mu_\star}$ which could lead to a deteriorated sampling quality. To better illustrate this phenomenon, we represent in Figure 2a a synthetic experiment, where $S_{\mu_\theta}$ does not recover $S_{\mu_\star}$ (by slightly shifting the mean of two modes after training the WGAN). In this setting, we clearly see in Figure 2b that Monte-carlo based methods worsen the WGAN. when $S_{\mu_\star} \not\subset S_{\mu_\theta}$, density ratios focus on local information and lead to non-optimal solutions. On the opposite, our method suggests to learn the optimal re-weighting of mass within the support $S_{\mu_\theta}$. Interestingly, on this synthetic dataset, it significantly reduces the Wasserstein distance to $\mu_\star$, see Figure 2c.

**4) Non-optimal discriminators.** Knowing that optimal discriminators would lead to non-optimal objectives (very low acceptance probabilities), previous approaches made sure that their obtained classifier is sufficiently far from the optimal classifier (Section 3.2 in (Azadi et al., 2019)). Authors have thus come up with heuristics to approximate density ratios: for example Azadi et al. (2019) fine tune a regularized discriminator, Grover et al. (2019) use a pre-trained neural network on ImageNet classification and only fine-tune the final layers for the binary classification task. In our method, on the contrary, we are still looking for the discriminator maximizing the Integral Probability Metric (Müller, 1997) in equation 3, linked to optimal transport.

## 4 EXPERIMENTS

In the following section, we illustrate the efficiency of the proposed methods, latentRS and latentGA on synthetic datasets. Then, we compare their performances with previous works on image datasets. On this image generation tasks, we empirically stress that both latentRS or latentGA methods slightly surpass density ratios based methods while significantly reducing the time complexity.

### 4.1 EVALUATION METRICS

To measure performances of GANs when dealing with low dimensional applications - as with synthetic datasets - we equip our space with the standard Euclidean distance. However, for high dimensional applications such as image generation, Brock et al. (2019); Kynkäänniemi et al. (2019) have shown that embedding images into a feature space with a pre-trained convolutional classifier provides more semantic information. In this setting, we consequently use the euclidean distance between the images' embeddings from a classifier. For a pair of images $(a, b)$, we define the distance $d(a, b)$ as $d(a, b) = \|\phi(a) - \phi(b)\|_2$ where $\phi$ is a pre-softmax layer of a supervised classifier, trained specifically on each dataset. Doing so, they will more easily separate images sampled from the target distribution $\mu_\star$ from the ones sampled by the distribution $\mu_\theta$.

We compare the performance of the different methods with a panel of evaluation metrics. To begin with, we use the Improved Precision/Recall (Improved PR) metric (Kynkäänniemi et al., 2019), a more robust version of the Precision/Recall metric which was first applied to the context of GANs by Sajjadi et al. (2018). The Improved PR metric is based on a non-parametric estimation of the support of both generated and real distributions using k-Nearest Neighbors. Besides, we also report two well-known metrics: the Earth Mover's Distance (EMD), the discrete version of the Wasserstein distance, and the average Hausdhorff distance (Hausd). EMD is a distance between probability distributions while Hausd. focuses on support estimations. These two measures are particularly interesting in GANs since one can compute them with collections of discrete points. Let $(x_1, \ldots, x_n)$ and $(y_1, \ldots, y_n)$ be two collections of $n$ data points and $\mathscr{S}$ be the set of permutations of $[1, n]$, then:

$$\text{EMD}(X, Y) = \min_{\sigma \in \mathscr{S}} \sum_{i=1}^{n} \|x_i, y_{\sigma_i}\| \quad \text{and} \quad \text{average Hausd}(X, Y) = \frac{1}{n} \sum_{x_i \in X} \min_{y_j \in Y} \|x_i, y_j\|$$

Besides, we argue that the Wasserstein distance should be a metric of reference when evaluating WGANs since it is directly linked to their objective function. Finally, for completeness, we report FID Heusel et al. (2017).

### 4.2 MIXTURE OF GAUSSIANS

Further experiments were ran on synthetic datasets with mixtures of 2D Gaussians, with either 4, 9, 16 or 25 components. When dealing with 2D mixtures of Gaussians, we used MLP with 4 hidden layers of 30 nodes for the generator, the discriminator and the importance weighter. As expected in this setting, a standard WGAN-GP combined with a connected latent space (*i.e.* multivariate normal or uniform), necessarily generates samples in-between two modes. Both Figure 1 and Figure 2a have stressed how the importance weighter can truncate latent space areas that are mapped outside the real data manifold and improve the EMD metric. More figures and details of the different evaluation metrics are given in Appendix.

### 4.3 IMAGE DATASETS

**MNIST, F-MNIST and Stacked MNIST.** We further study the efficiency of the proposed methods on three image datasets: MNIST (LeCun et al., 1998), FashionMNIST (F-MNIST) (Xiao et al., 2017), and Stacked MNIST (Metz et al., 2016) a highly disconnected datasets with 1,000 classes. For MNIST, F-MNIST and Stacked MNIST, we follow Khayatkhoei et al. (2018) and use a standard CNN architecture composed of a sequence of 3x3 convolution layer, relu activation with nearest neighbor upsampling. To exhibit the efficiency of the proposed methods in different settings, we use hinge loss with gradient penalty (Hinge-GP) (Miyato et al., 2018) on MNIST and F-MNIST, and a Wasserstein loss with gradient penalty (Gulrajani et al., 2017) on Stacked Mnist. For the importance weighter $w_\varphi$, we use an MLP architecture with fully-connected layers and relu activation. $w_\varphi$ has 4 hidden layers, each having its width four times larger than the dimension of the latent space.

| **MNIST** | Prec. (↑) | Rec. (↑) | EMD (↓) | Hausd (↓) | FID (↓) | Inference |
|---|---|---|---|---|---|---|
| Hinge-GP | $87.4_{\pm0.9}$ | $94.6_{\pm0.4}$ | $24.9_{\pm0.3}$ | $21.9_{\pm0.2}$ | $53.6_{\pm7.2}$ | 0.9 |
| HGP: SIR-GAN | $89.4_{\pm0.5}$ | $94.3_{\pm0.5}$ | $24.1_{\pm0.1}$ | $21.2_{\pm0.2}$ | $\mathbf{35.9_{\pm3.1}}$ | 52.0 |
| HGP: DOT | $89.5_{\pm0.6}$ | $94.0_{\pm0.3}$ | $24.8_{\pm0.2}$ | $21.7_{\pm0.2}$ | $43.3_{\pm3.4}$ | 43 |
| HGP: latentRS (⋆) | $88.9_{\pm0.4}$ | $\mathbf{94.7_{\pm0.7}}$ | $24.2_{\pm0.3}$ | $21.4_{\pm0.2}$ | $37.3_{\pm3.2}$ | 2.0 |
| HGP: latentGA (⋆) | $\mathbf{91.4_{\pm1.0}}$ | $92.9_{\pm0.4}$ | $\mathbf{23.5_{\pm0.1}}$ | $\mathbf{19.8_{\pm0.2}}$ | $38.2_{\pm3.8}$ | 18 |
| **F-MNIST** | | | | | | |
| Hinge-GP | $86.4_{\pm0.6}$ | $86.8_{\pm0.6}$ | $68.6_{\pm0.4}$ | $58.4_{\pm0.3}$ | $598.9_{\pm55.5}$ | 0.9 |
| HGP: SIR-GAN | $86.8_{\pm0.4}$ | $87.4_{\pm0.5}$ | $68.9_{\pm0.8}$ | $57.4_{\pm0.3}$ | $558.9_{\pm58.7}$ | 52.0 |
| HGP: DOT | $88.2_{\pm0.6}$ | $85.8_{\pm0.5}$ | $70.4_{\pm0.7}$ | $57.0_{\pm0.2}$ | $797.5_{\pm84.9}$ | 43 |
| HGP: latentRS (⋆) | $86.8_{\pm0.8}$ | $\mathbf{87.5_{\pm0.9}}$ | $67.6_{\pm0.6}$ | $58.6_{\pm0.5}$ | $\mathbf{438.3_{\pm50.2}}$ | 2.1 |
| HGP: latentGA (⋆) | $\mathbf{88.4_{\pm0.7}}$ | $86.8_{\pm0.7}$ | $\mathbf{67.0_{\pm0.9}}$ | $\mathbf{56.4_{\pm0.2}}$ | $475.5_{\pm58.5}$ | 18 |
| **Stacked MNIST** | | | | | | |
| WGAN-GP | $88.7_{\pm0.8}$ | $87.3_{\pm0.5}$ | $81.1_{\pm0.2}$ | $75.7_{\pm0.2}$ | $594.2_{\pm8.2}$ | 0.9 |
| WGP: SIR-GAN | $89.9_{\pm0.5}$ | $88.4_{\pm0.6}$ | $80.3_{\pm0.2}$ | $75.1_{\pm0.1}$ | $528.1_{\pm13.4}$ | 52.0 |
| WGP: DOT | $\mathbf{90.8_{\pm0.5}}$ | $\mathbf{90.2_{\pm0.7}}$ | $\mathbf{79.4_{\pm0.1}}$ | $\mathbf{74.1_{\pm0.2}}$ | $\mathbf{484.0_{\pm8.8}}$ | 43 |
| WGP: latentRS (⋆) | $90.4_{\pm0.4}$ | $86.6_{\pm0.5}$ | $80.3_{\pm0.1}$ | $75.3_{\pm0.1}$ | $568.5_{\pm9.1}$ | 1.9 |
| WGP: latentGA (⋆) | $90.3_{\pm0.7}$ | $87.2_{\pm0.5}$ | $80.1_{\pm0.2}$ | $74.9_{\pm0.2}$ | $530.1_{\pm6.6}$ | 18 |

Table 1: On the image generation task, latentGA seems the best performer and latentRS matches sato with a significantly reduced inference cost (by an order of 20). $\pm$ is 97% confidence interval. Inference refers to the time needed to compute one image on a NVIDIA V100 GPU.

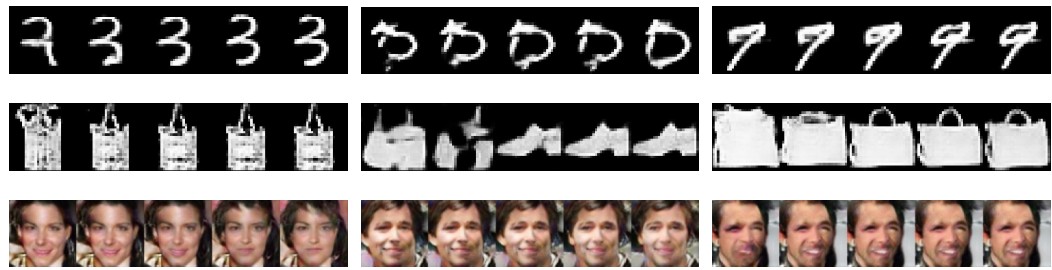

Figure 3: Gradient ascent on latent importance weights (latentGA): the quality is gradually improved as we move to higher importance weights. Each image is generated only for the purpose of visualization, and one can run this gradient ascent directly in the latent space with the importance weighter.

For exhaustivity, we compare latentRS and latentGA with previous works leveraging density ratios. In particular, we implemented a wide set of post-processing methods for GANs: DRS (Azadi et al., 2019), MH-GAN (Turner et al., 2019), SIR-GAN (Grover et al., 2019) and DOT (Tanaka, 2019). Similarly to Azadi et al. (2019), we take the discriminator at the end of the adversarial training, fine-tune it with the binary cross-entropy loss and select the best model in terms of EMD. During fine-tuning, we keep the gradient penalty or spectral normalization, otherwise the classifier easily separates real from generated data, which leads to a degraded performance, as shown in Figure 2a. Following Azadi et al. (2019); Grover et al. (2019), we do not include explicit mechanism to calibrate the classifier. To the extent of our knowledge, we are the first to empirically compare such a wide variety of Monte-Carlo methods on different datasets and metrics.

The main results of this comparison are shown in Table 1 (see Appendix for more details). We see that, except for Stacked MNIST, both of our methods outperform every other method on precision, av. Hausdhorff and the EMD metric. Interestingly, latentGA seems to be the strongest one. In Figure 3, we show how samples evolve when performing latent gradient ascent on the importance weights. As expected, as importance weights are increased, the quality of the generated images significantly improves. Besides, a strong contribution of the paper also resides in notably speeding-up the inference procedure. As shown in Table 1, the inference time of a given data point is 25 times faster with latentRS than with SIR-GAN.

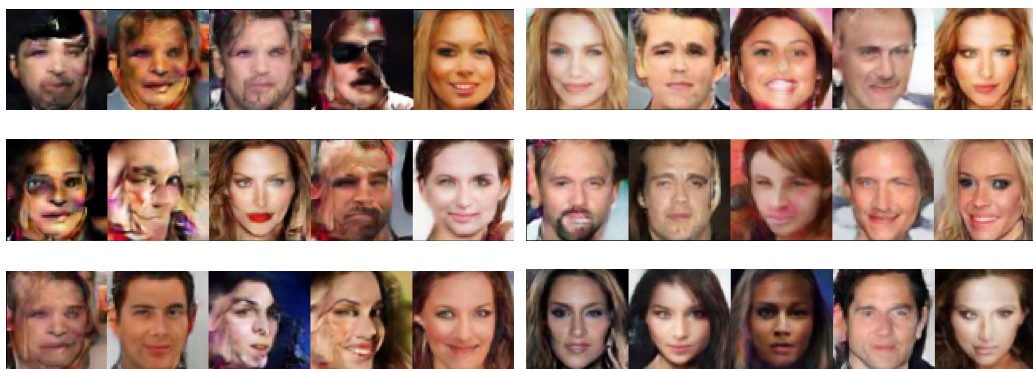

Figure 4: Ranking CelebA faces with the trained importance weighter. For each line, we compare the worst-5% vs best-5% of the generated samples. We see a significant difference in terms of quality.

**CelebA** is a large-scale dataset of faces covering a variety of poses. We train the models at 64x64 resolution. Following recent studies (Brock et al., 2019), the discriminator is trained with the hinge loss and spectral normalization (Miyato et al., 2018). For the generator network, residual connections (He et al., 2016) are used alongside self-modulation layers (Chen et al., 2019). The importance weighter is a simple 4 hidden-layer MLP with a width 10 times larger than the latent space dimension.

In this one-class high-dimensional dataset, the importance weighter still managed to learn some meaningful features. First, Figure 3 higlights a subtle improvement of the generated images when performing latentGA. Second, when ranking generated images with the importance weighter and comparing the top-5% vs worst-5% in Figure 4, we observe some differences in quality. However, on a broader scale, the importance weighter does not bring a clear improvement on neither the EMD nor the Hausdhorff metric. Interestingly, this is also the case for any of the different rejection methods (see Appendix for details).

We argue that in this one-class generation task, post-processing the generated samples is not as efficient as in a multi-modal setting (*e.g.* MNIST, FMNIST, Stacked MNIST). Intuitively, it is a much easier task to remove generated samples that are out of the target manifold than to discriminate between samples that already share similarities with training samples. It further stresses that this family of methods is useful if one needs to insert disconectedness in the modeled distribution. However, when the target distribution is a single-class distribution with a connected support, their efficiency decrease. To illustrate this, we added in Appendix a figure highlighting samples generated by a trained WGAN on Celeba 64x64, ranked by the discriminator. We observe that on these images, the discriminator does not correlate well with human judgement prohibiting the importance weighter to learn a meaningful signal.

## 5 Conclusion

In this paper, we provide insights on improving the learning of disconnected manifolds with GANs. Given the existence of the no GAN's land, latent space areas mapping outside the target manifold, we provide two methods to truncate them. Contrary to previous works focusing on learning density ratios in the output space, both of our methods are based on training adversarially a neural network learning importance weights in the latent space. On the task of image generation, both of the proposed methods were shown to be empirically efficient while significantly reducing the inference time (latentRS by an order of 20), when compared to density ratio based methods. This paper has specifically stressed the efficiency of post-training methods when dealing with highly disconnected target distributions.

However, when dealing with single-class connected distributions or class-conditioning generative models, the efficiency of such methods is not clear. We argue that one of the reason is that, once the generator maps correctly inside the target manifold, it is a much harder task to discriminate between realistic and fake samples. A potential future work would therefore be to investigate how can we help the discriminator better classify among the set of generated images.

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

## A   EVALUATION DETAILS

**Precision recall metric.**   For the precision-recall metric, we use the algorithm from Khayatkhoei et al. (2018). Namely, when comparing the set of real data points $(x_1, ..., x_n)$ with the set of fake data points $(y_1, ..., y_n)$:

A point $x_i$ has a recall $r(x_i) = 1$ if there exists $y_j$, such that $\|x_i - y_j\| \leq \|y_j - y_j(k)\|$, where $y_j(k)$ is the k-nearest neighbor of n. Finally, the recall is the average of individual recall: $\frac{1}{n} \sum_i r(x_i)$.

A point $y_i$ has a precision $p(y_i) = 1$ if there exists $x_j$, such that $\|y_i - x_j\| \leq \|x_j - x_j(k)\|$, where $x_j(k)$ is the k-nearest neighbor of n. Finally, the precision is the average of individual precision: $\frac{1}{n} \sum_i p(x_i)$.

**Images' embeddings.**   As mentioned earlier, for images we use the distance between embeddings of images in a neural network trained specifically for classification on this dataset. For Stacked Mnist, we use a MNIST classifier on each output channel and simply stack the three embedding vectors.

**Parameters.**   For all datasets, we use $k = 3$ (3rd nearest neighbor). For MNIST, F-MNIST and Stacked MNIST, we use a set of $n = 2048$ points. For CelebA, we use a set of $n = 1024$ points. This is also valid for the other metrics used: EMD, Av. Hausd. For FID on CelebA, we use the standard protocol evaluation with Inception Net and 50k data points.

## B   HYPER-PARAMETERS.

**SIR:** Model selection: we fine-tune with a binary cross-entropy loss the discriminator from the end of the adversarial training and select the best model in terms of EMD.

We use then use Sampling-Importance-Resampling algorithm, with sets of n=40 points.

**DRS:** Model selection: we fine-tune with a binary cross-entropy loss the discriminator from the end of the adversarial training and select the best model in terms of EMD.

We use the standard Rejection Sampling algorithm, without artificially increasing the acceptance rate such as Azadi et al. (2019). We use regularized discriminator (with gradient penalty or spectral normalization), which avoids the acceptance rate falling to almost zero.

**MH-GAN:** Model selection: we fine-tune with a binary cross-entropy loss the discriminator from the end of the adversarial training and select the best model in terms of EMD. We use the independance Metropolis-Hastings algorithm with Markov Chains of 40 points, and select the last point.

**DOT:** Model selection: we fine-tune with the dual wasserstein loss the discriminator from the end of the adversarial training and select the best model in terms of EMD. We then perform a projected gradient descent as described in Tanaka (2019) with SGD, with $N_{steps} = 10$ and $\varepsilon = 0.01$.

**LRS:** For MNIST, F-MNIST and Stacked MNIST, we use the same hyper-parameters: $\lambda_1 = 10$, $\lambda_2 = 2$ and $m = 3$. $w_\varphi$ is a standard MLP with 4 hidden layers, each having 600 nodes (6x dimension of latent space), and relu activation. The output layer is 1-dimensional and with a relu activation.

For CelebA, we use: $\lambda_1 = 50$, $\lambda_2 = 2$ and $m = 3$. $w_\varphi$ is a standard MLP with 4 hidden layers, each having 1280 nodes (10x dimension of latent space), and relu activation. The output layer is 1-dimensional and with a relu activation.

For the adversarial training of importance weights, we use the discriminator from the end of the standard adversarial training (generator vs discriminator). We then alternate between 1 step of $w_\varphi$ and 1 update of $D_\alpha$.

**LGA:** We use the same neural network than in LRS. The hyper-parameters for this method are similar to DOT: number of steps of gradient ascent $N_{steps}$ and learning rate $\varepsilon$. We choose $N_{steps} = 10$ and $\varepsilon = 0.1$.

## C    VISUALIZATION AND RESULTS FOR SYNTHETIC DATASETS

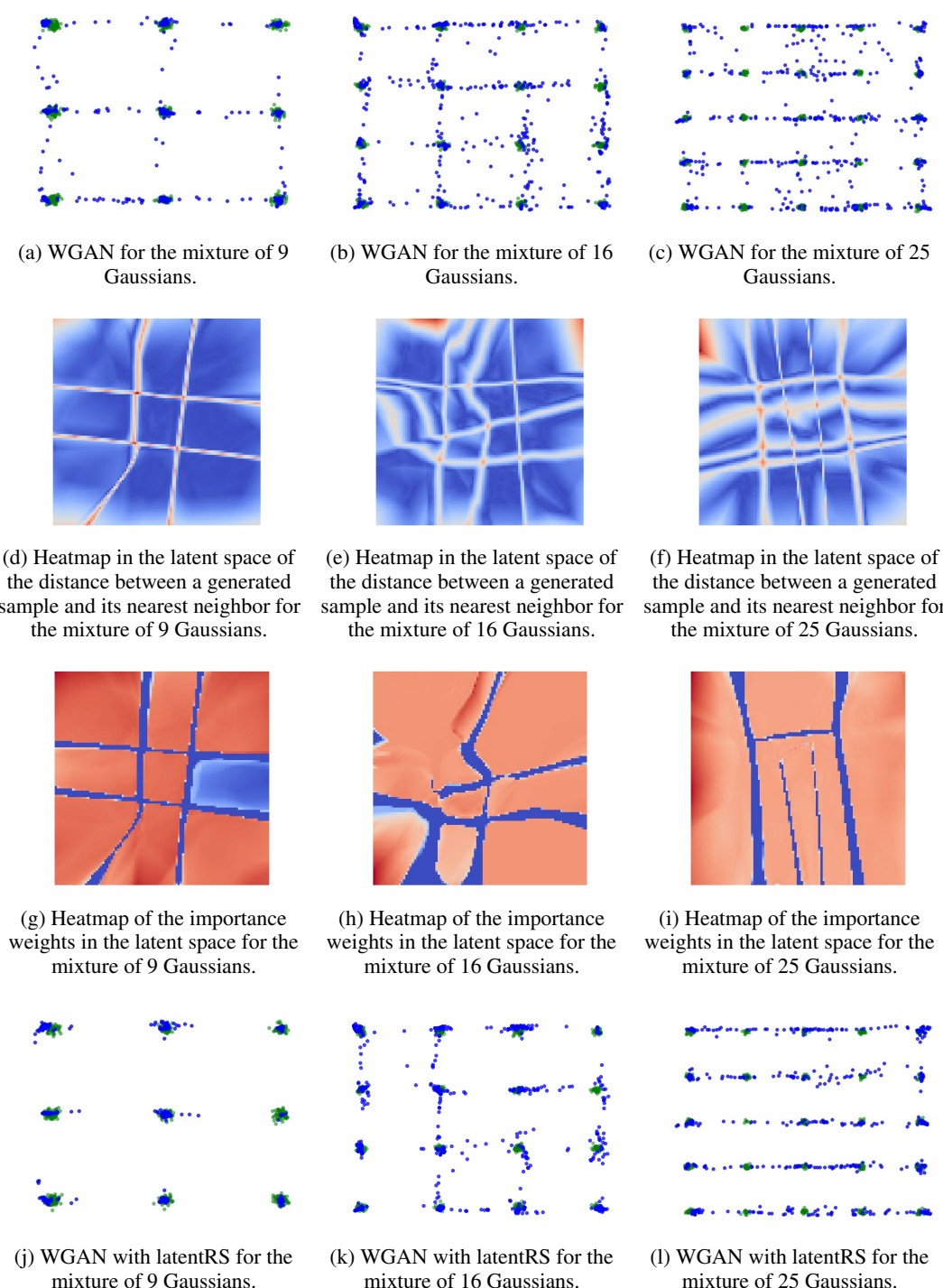

(a) WGAN for the mixture of 9 Gaussians.

(b) WGAN for the mixture of 16 Gaussians.

(c) WGAN for the mixture of 25 Gaussians.

(d) Heatmap in the latent space of the distance between a generated sample and its nearest neighbor for the mixture of 9 Gaussians.

(e) Heatmap in the latent space of the distance between a generated sample and its nearest neighbor for the mixture of 16 Gaussians.

(f) Heatmap in the latent space of the distance between a generated sample and its nearest neighbor for the mixture of 25 Gaussians.

(g) Heatmap of the importance weights in the latent space for the mixture of 9 Gaussians.

(h) Heatmap of the importance weights in the latent space for the mixture of 16 Gaussians.

(i) Heatmap of the importance weights in the latent space for the mixture of 25 Gaussians.

(j) WGAN with latentRS for the mixture of 9 Gaussians.

(k) WGAN with latentRS for the mixture of 16 Gaussians.

(l) WGAN with latentRS for the mixture of 25 Gaussians.

Figure 5: Synthetic examples on mixtures of Gaussians with respectively (per column) 9, 16 and 25 components. Real samples in green and generated points in blue.

| Mixture of 4 Gaussians | Prec. | Rec. | EMD | Hausd. |
|---|---|---|---|---|
| WGAN-GP | $95.0_{\pm 0.5}$ | $78.5_{\pm 1.1}$ | $0.50_{\pm 0.05}$ | $0.85_{\pm 0.05}$ |
| WGAN-GP with latentRS ($\star$) | $99.2_{\pm 0.1}$ | $70.8_{\pm 2.2}$ | $0.37_{\pm 0.07}$ | $0.10_{\pm 0.02}$ |
| **Mixture of 9 Gaussians** | | | | |
| Hinge-GP | $86.3_{\pm 0.3}$ | $76.2_{\pm 0.8}$ | $0.45_{\pm 0.07}$ | $0.57_{\pm 0.07}$ |
| HGP: latentRS ($\star$) | $91.0_{\pm 0.3}$ | $73.0_{\pm 1.4}$ | $0.40_{\pm 0.06}$ | $0.23_{\pm 0.04}$ |
| **Mixture of 16 Gaussians** | | | | |
| Hinge-GP | $70.6_{\pm 0.3}$ | $68.2_{\pm 0.8}$ | $1.10_{\pm 0.07}$ | $0.85_{\pm 0.02}$ |
| HGP: latentRS ($\star$) | $74.0_{\pm 0.3}$ | $69.5_{\pm 1.4}$ | $1.00_{\pm 0.04}$ | $0.54_{\pm 0.04}$ |
| **Mixture of 25 Gaussians** | | | | |
| Hinge-GP | $63.4_{\pm 0.3}$ | $64.6_{\pm 0.8}$ | $1.35_{\pm 0.08}$ | $1.10_{\pm 0.07}$ |
| HGP: latentRS ($\star$) | $70.6_{\pm 0.3}$ | $67.0_{\pm 1.4}$ | $1.25_{\pm 0.05}$ | $0.8_{\pm 0.04}$ |

Table 2: Performance of the latentRS method on the estimation of 2D mixtures of Gaussians with respectively 4, 9, 16 or 25 components. Each mode has a variance $\sigma^2 = 0.025$ is separated by a distance $d = 5$. As expected the latentRS method removes data points that are off the target manifold and therefore significantly improves the Hausdhorff metric.

# D   COMPARISONS WITH CONCURRENT METHODS ON SYNTHETIC AND REAL-WORLD DATASETS

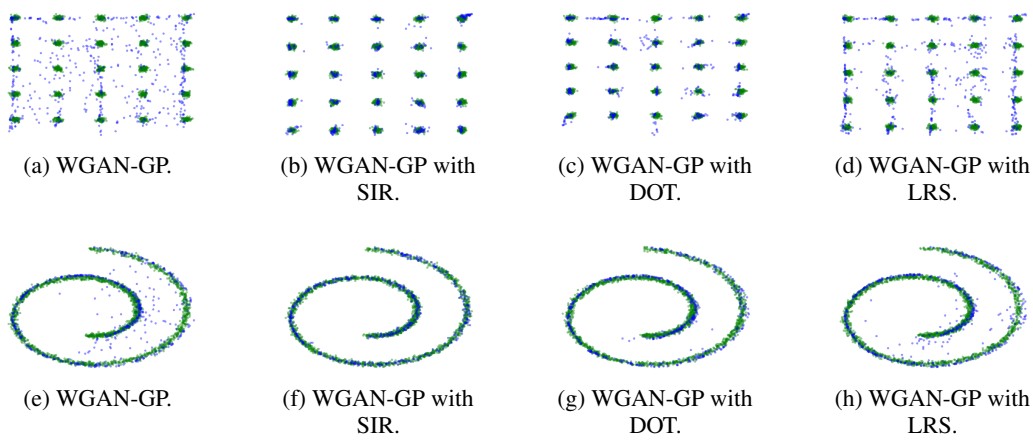

(a) WGAN-GP.    (b) WGAN-GP with SIR.    (c) WGAN-GP with DOT.    (d) WGAN-GP with LRS.

(e) WGAN-GP.    (f) WGAN-GP with SIR.    (g) WGAN-GP with DOT.    (h) WGAN-GP with LRS.

Figure 6: Comparisons of concurrent methods on synthetic datasets: mixtures of 25 Gaussians and on 2D Swiss Roll. Experimental setting from Tanaka (2019). Real samples in green and generated points in blue.

| Mixture of 25 Gaussians | Prec. | Rec. | EMD |
|---|---|---|---|
| WGAN-GP | 24.7±0.5 | 90.4±1.0 | 0.044±0.001 |
| WGAN-GP with DRS | **86.9**±0.6 | 84.0±0.5 | 0.038±0.002 |
| WGAN-GP with SIR | 84.3±0.8 | **90.0**±0.7 | 0.041±0.001 |
| WGAN-GP with DOT | 72.7±1.0 | 76.3±0.8 | **0.035**±0.002 |
| WGAN-GP with latentRS (⋆) | 36.2±0.8 | 88.5±0.9 | 0.036±0.001 |
| **Swiss Roll** | | | |
| WGAN-GP | 77.9±0.5 | 90.3±0.7 | 0.030±0.002 |
| WGAN-GP: DRS | **97.7**±0.2 | **94.4**±0.5 | 0.036±0.004 |
| WGAN-GP: SIR | 97.3±0.6 | 93.9±0.7 | 0.037±0.003 |
| WGAN-GP: DOT | 92.9±0.6 | 77.7±0.5 | 0.029±0.003 |
| WGAN-GP: latentRS (⋆) | 84.3±0.8 | 90.1±0.8 | **0.025**±0.002 |

Table 3: Comparison of latentRS with concurrent methods on two synthetic datasets in the same setting as Tanaka (2019). Our method enables a consistent gain in EMD, surpassing other methods on Swiss Roll and slightly behind DOT on Mixture of 25 Gaussians.

| MNIST | Prec. (↑) | Rec. (↑) | EMD (↓) | av. Hausd (↓) | FID (↓) |
|---|---|---|---|---|---|
| Hinge-GP | 87.4±0.9 | 94.6±0.4 | 24.9±0.3 | 21.9±0.2 | 53.6±7.2 |
| HGP: DRS | 89.0±0.5 | 94.4±0.5 | 24.3±0.3 | 21.2±0.2 | 41.3±8.2 |
| HGP: SIR-GAN | 89.4±0.5 | 94.3±0.5 | 24.1±0.1 | 21.2±0.2 | **35.9**±3.1 |
| HGP: MH-GAN | 89.4±0.6 | 94.5±0.3 | 24.4±0.3 | 21.1±0.1 | 43.2±5.7 |
| HGP: DOT | 89.5±0.6 | 94.0±0.3 | 24.8±0.2 | 21.7±0.2 | 43.3±3.4 |
| HGP: latentRS (⋆) | 88.9±0.4 | **94.7**±0.7 | 24.2±0.3 | 21.4±0.2 | 37.3±3.2 |
| HGP: latentGA (⋆) | **91.4**±1.0 | 92.9±0.4 | **23.5**±0.1 | **19.8**±0.2 | 45.2±3.8 |
| **F-MNIST** | | | | | |
| Hinge-GP | 86.4±0.6 | 86.8±0.6 | 68.6±0.4 | 58.4±0.3 | 598.9±55.5 |
| HGP: DRS | 86.7±0.7 | 87.4±0.5 | 68.7±0.7 | 57.7±0.2 | 546.8±37.3 |
| HGP: SIR-GAN | 86.8±0.4 | 87.4±0.5 | 68.9±0.8 | 57.4±0.3 | 558.9±58.7 |
| HGP: MH-GAN | 87.8±0.5 | 87.0±0.4 | 68.8±0.6 | 57.6±0.2 | 555.3±42.4 |
| HGP: DOT | 88.2±0.6 | 85.8±0.5 | 70.4±0.7 | 57.0±0.2 | 797.5±84.9 |
| HGP: latentRS (⋆) | 86.8±0.8 | **87.5**±0.9 | 67.6±0.6 | 58.6±0.5 | **438.3**±50.2 |
| HGP: latentGA (⋆) | **88.4**±0.7 | 86.8±0.7 | **67.0**±0.9 | **56.4**±0.2 | 475.5±58.5 |
| **Stacked MNIST** | | | | | |
| WGAN-GP | 88.7±0.8 | 87.3±0.5 | 81.1±0.2 | 75.7±0.2 | 594.2±8.2 |
| WGP - DRS | 89.9±0.5 | 88.5±0.6 | 80.2±0.1 | 75.1±0.1 | 527.2±9.2 |
| WGP: SIR-GAN | 89.9±0.5 | 88.4±0.6 | 80.3±0.2 | 75.1±0.1 | 528.1±13.4 |
| WGP: MH-GAN | 90.3±0.6 | 88.9±0.5 | 80.2±0.1 | 75.1±0.1 | 527.2±13.4 |
| WGP: DOT | **90.8**±0.5 | **90.2**±0.7 | **79.4**±0.1 | **74.1**±0.2 | **484.0**±8.8 |
| WGP: latentRS (⋆) | 90.4±0.4 | 86.6±0.5 | 80.3±0.1 | 75.3±0.1 | 568.5±9.1 |
| WGP: latentGA (⋆) | 90.3±0.7 | 87.2±0.5 | 80.1±0.2 | 74.9±0.2 | 530.1±6.6 |
| **CelebA 64x64** | | | | | |
| Hinge-SN | **88.2**±0.6 | 66.7±0.8 | 37.7±0.2 | 34.5±0.2 | **17.8**±0.1 |
| HSN: DRS | 88.1±0.7 | 66.5±1.2 | 38.3±0.7 | 34.5±0.2 | 19.1±0.2 |
| HSN: SIR | 87.9±0.6 | 66.9±1.3 | 38.1±0.2 | 34.4±0.2 | 19.2±0.2 |
| HSN: MH | 88.1±0.8 | 67.3±1.3 | 38.1±0.3 | 34.4±0.3 | 19.2±0.2 |
| HSN: DOT | 88.0±0.8 | 67.4±1.5 | 37.4±0.3 | 34.3±0.3 | 18.0±0.1 |
| HSN: latentRS (⋆) | 88.0±0.6 | **68.1**±1.5 | **37.3**±0.3 | **34.1**±0.3 | 18.0±0.1 |
| HSN: latentGA (⋆) | 88.1±0.6 | 66.8±1.1 | 37.4±0.2 | 34.3±0.2 | 18.1±0.2 |

Table 4: Comparing latentRS and latentGA with previous works (DRS, MH-GAN, SIR-GAN, DOT) on the image generation task. Interestingly, latentGA seems the best performer and latentRS matches sato with a significantly reduced inference cost (by an order of 20). ± is 97% confidence interval. Inference refers to the time needed to compute one image on a NVIDIA V100 GPU. ± is 97% confidence interval.

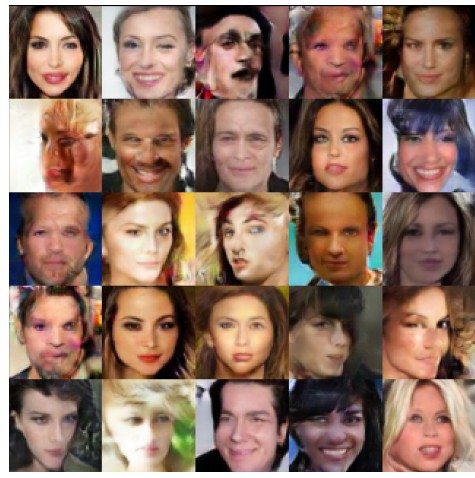

(a) Generated samples on the Celeba dataset 64x64 ordered by the discriminator (left to right, top to bottom).

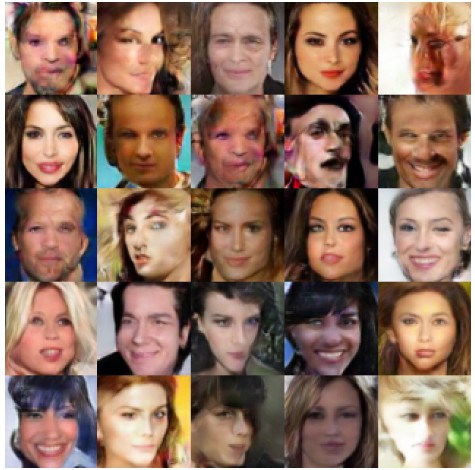

(b) Generated samples on the Celeba dataset 64x64 ordered by the importance weighter (left to right, top to bottom).

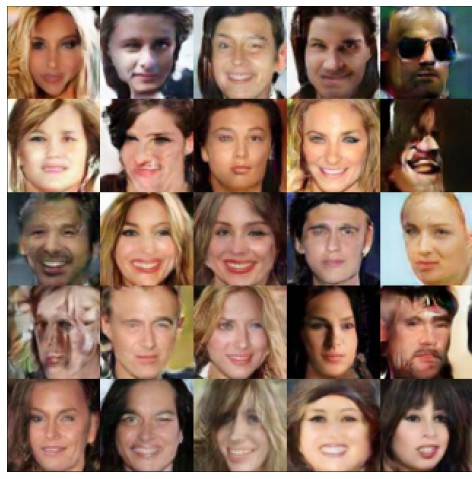

(c) Generated samples on the Celeba dataset 64x64 ordered by the discriminator (left to right, top to bottom).

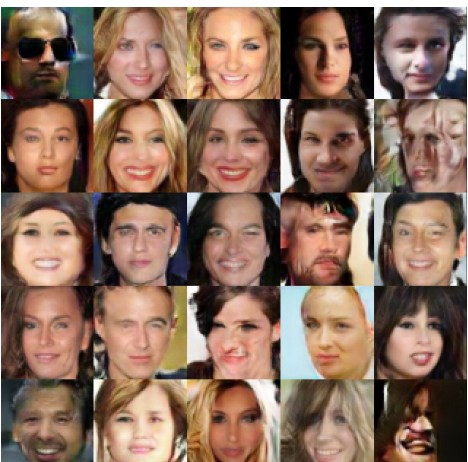

(d) Generated samples on the Celeba dataset 64x64 ordered by the importance weighter (left to right, top to bottom).

Figure 7: We observe that the ranking with the discriminator does not perfectly correlate with human judgment, prohibiting the importance weighter to learn meaningful features. Consequently, the target distribution is not better approximated.

