# OpenReview forum: "Learning Disconnected Manifolds: Avoiding The No Gan's Land by Latent Rejection"
_ICLR.cc/2021/Conference — Reject_

### Official Review · AnonReviewer4 · 2020-10-27
**a straightforward amortization of the discriminator outputs with a limited empirical study**

**Rating:** 4
**Confidence:** 4

**Review:**

The paper proposes a method for an improvement of generative adversarial models via post-processing its latent variable distribution. To be more precise, the method proposes to train an additional neural network that outputs an important weight for each point of the latent space, thus reweighting the final distribution in the space of images. For the optimization of this network, the authors use the dual form of the Wasserstein distance, where they multiply the initial latent density by the output of the network. To fix the ill-behaved objective, the authors add two regularization terms to it. The proposed objective is then validated on 3 MNIST-like datasets quantitatively and on CelebA qualitatively.

Review:
My major concern is the limited theoretical novelty together with modest empirical study. Let me clarify. I think the idea to put the filtering stage into the latent space is indeed worthy. However, the straightforward amortization of the discriminator network via a fully connected network is challenging due to the described computational problems and usually high dimensionality of the latent space. Furthermore, the verification of the method on MNIST-like data does not seem convincing, especially when the relevant works provide a comparison on ImageNet (Azadi 2018, Neklyudov 2019).

Additional comments:
1. perhaps, I'm missing something, but for me, it is not clear why the objective in equation (3) corresponds to the optimization of Wasserstein distance in the space of images w.r.t. the parameters alpha and phi. I mean that there are even no guarantees that \widehat{\gamma} is a distribution.
2. "since the rejection sampling scheme is now tractable, we do not need to implement the MH algorithm or the importance sampling". Firstly, I do not understand why the rejection sampling is tractable. The regularization term does not provide any guarantees for the maximum value of the density ratio. Secondly, even if the rejection sampling is tractable, I still find the MH algorithm more efficient: it does not require the evaluation of the constant; given the same proposal, MH's acceptance rate is greater or equal to the acceptance rate of the rejection sampling.
3. the authors claim that reweighting in the latent space allows for better support coverage than the methods operating on the image space. Although I believe that such an effect occurs, I wouldn't expect the quality of images to be high. Indeed, this additional coverage could be produced by sampling from the low-density regions of the latent distribution. It is clear that such regions are underrepresented during the training. Moreover, there is empirical evidence of the deteriorating quality of images for latent distributions with higher variance (see Brock 2018).
4. the bottom of page 3. DRS does not assure sampling from the target distribution since it adjusts the constant and uses an approximation of density ratio. In contrast, the MH algorithm provides some guarantees by upper bounding the total variation distance between the stationary distribution and the target (see Neklyudov 2019).

minor comments:
1. abstract. I would suggest finding a better analog for the phrase "inject disconnectedness". It does not sound like a desirable feature of your model when we speak about GANs, especially at the beginning of the paper, where few context is given. I would propose something like "postselection" or "filtering".
2. eq. 4, the signs of regularization terms are incorrect
3. typo on page 5, item 2). every methods -> every method

References:
1. (Azadi 2018) Azadi, Samaneh, Catherine Olsson, Trevor Darrell, Ian Goodfellow, and Augustus Odena. "Discriminator rejection sampling." arXiv preprint arXiv:1810.06758 (2018).
2. (Brock 2018) Brock, Andrew, Jeff Donahue, and Karen Simonyan. "Large scale gan training for high fidelity natural image synthesis." arXiv preprint arXiv:1809.11096 (2018).
3. (Neklyudov 2019) Neklyudov, Kirill, Evgenii Egorov, and Dmitry P. Vetrov. "The Implicit Metropolis-Hastings Algorithm." In Advances in Neural Information Processing Systems, pp. 13954-13964. 2019.

---

> ### Author Response · Authors · 2020-11-16
> **Authors response to AnonReviewer4**
>
> First, we would like to thank you for reviewing our paper and for your valuable comments. We will try to answer your main concerns.
>
> 1) "It is not clear why the objective in equation (3) corresponds to the optimization of Wasserstein distance in the space of images w.r.t. the parameters alpha and phi. I mean that there are even no guarantees that $\widehat{\gamma}$ is a distribution."
> First, regarding $\widehat{\gamma}$, we add a strong regularization term forcing $\widehat{\gamma}$ to define a normalized distribution (self-normalization). We track the quantity $(\{E}_{z\sim Z} w_\varphi (z) -1)^2$ during training, and it is indeed 0 after a few training steps, which gives us a guarantee that $\widehat{\gamma}$ defines a distribution.
> From here, it is clear that we are minimizing the Wasserstein distance. The Wasserstein distance consists in estimating the difference of two integrals. On the left side (target distribution), we do not bring any modification to the standard setting. On the right side (generated distribution), the generated distribution is modified by the learnt importance weights, which is a very classical way of performing importance sampling.
>
> 2) "Firstly, I do not understand why the rejection sampling is tractable. The regularization term does not provide any guarantees for the maximum value of the density ratio."
> You are right, if the regularization term fails, the constant can be higher than what we wanted. However, we use strong penalty terms (high $\lambda_2$) and did not empirically observe a constant higher than the maximum that we set.
>
> 3) "Secondly, even if the rejection sampling is tractable, I still find the MH algorithm more efficient: it does not require the evaluation of the constant."
> Actually, as explained in the Related Work section, the two algorithms are difficult to compare. On one side, Rejection Sampling ensures to sample the target distribution, with no guarantee on the computational budget; on the other side, Metropolis-Hastings only ensures asymptotic convergence, but with an explicit control of the computational budget. Indeed, in practice, Metropolis-Hastings is used by selecting the last sample of a Markov Chain.
> Moreover, one could also use the MH algorithm with the learnt latent importance weights. We empirically tried it and we saw no improvement over latentRS.
>
> 4) "DRS does not assure sampling from the target distribution since it adjusts the constant and uses an approximation of density ratio. In contrast, the MH algorithm provides some guarantees by upper bounding the total variation distance between the stationary distribution and the target (see Neklyudov 2019)."
> Yes, you are right, there is a typo there. We were comparing Rejection Sampling vs Metropolis-Hastings in their original form and assumptions, not in the context of GANs. As mentionned above, Rejection Sampling ensures to sample the target distribution, with no guarantee on the computational budget; while Metropolis-Hastings ensures asymptotic convergence, but with an explicit control of the computational budget (length of the Markov Chain).
> Moreover, we would like to point out that the assumptions of rejection sampling/Metropolis-Hastings are not fullfiled in the context of GANs generating submanifolds of a high-dimensional space. Indeed, in this case, the support of the target distribution is not included in the support of the modelled distribution (see [1]). That is why DRS/MH-GANs have flaws.
> [1] Arjovsky, Martin, and Leon Bottou. "Towards Principled Methods for Training Generative Adversarial Networks." ICLR 2017.
>
> Furthermore, as asked by AnonReviewer1, we have updated the paper with according experiments on Swiss Roll and Mixture of 25 Gaussians (Figure 6 and Table 3 in Appendix). It supports our point: latentRS surpasses or matches the other methods on EMD, which is an empirical measure of the Wasserstein distance between generated and target distributions. Methods based on density-ratios do not always improve the goodness of fit measured in terms of EMD.

---

> > ### Comment · AnonReviewer4 · 2020-11-21
> > **response to the authors**
> >
> > I have read the rebuttal. Unfortunately, my concerns are not addressed by the authors.
> >
> > 1,2. Adding the regularization terms to your objective does not provide any theoretical guarantees. Moreover, even your empirical check of these conditions is not sufficient to make such a claim since you are using stochastic estimates rather than exact integration over the space, which is impossible here.
> >
> > 3,4. Indeed, both algorithms are hard to compare directly. However, I do not find your informal comparison to be an argument for the usage of rejection sampling. Moreover, the distinction between computational budget and asymptotic behavior is not clear to me at all since they are tightly related. Indeed, you cannot say that the rejection sampling generates from the target distribution from the start since it could spend a lot of iterations (on average, more than MH with the same proposal) to accept a single point, as well as you can treat MH burn-in as computational expenses.
> >
> > 5. The authors do not address my concerns about MNIST-like datasets used for empirical comparison.
> > 6. The authors do not address my 3rd additional comment.
> >
> > Overall, I do not see any reasons to raise my score.

---

> > > ### Author Response · Authors · 2020-11-24
> > > **Authors response**
> > >
> > > 1,2. A neural network can be considered as a bounded Lipschitz function (e.g. a ReLU network is piece-wise linear). Thus, it comes from [1, Theorem 3.1] that the empirical measure $\frac{1}{n} \sum w_\varphi(z)$ converges to $E_{z\sim Z}[w_\varphi(z)]$. In our setting the consistency is an immediate consequence of theorems on Donsker's classes for the Lipschitz cases.
> > >
> > > [1] van der Vaart, Aad W., and Jon A. Wellner Wellner. "Empirical processes indexed by estimated functions." Asymptotics: particles, processes and inverse problems. Institute of Mathematical Statistics, 2007. 234-252.
> > >
> > > 3,4. We do not provide any new claim or comparison between these two algorithms, and were just presenting standard results for Monte-Carlo algorithms: Rejection Sampling ensures sampling from the target distribution, but the acceptance rate (1/M) can be very large in high-dimension; Metropolis-Hastings ensures asymptotic convergence. We refer to [2] for the proofs of these properties and more details.
> > >
> > > [2] Robert, Christian, and George Casella. Monte Carlo statistical methods. Springer Science & Business Media, 2013.

---

### Official Review · AnonReviewer1 · 2020-10-28
**Treating the problem in latent space is promising, but experiments are not convincing**

**Rating:** 4
**Confidence:** 4

**Review:**

This work aims at improving the sample quality of generative models through better sampling, which is a relevant problem and has brought about a line of work [1,2,3,4,5], to name a few. By leveraging the idea of importance sampling, the authors train an additional network. The latter uses the information contained in the learned discriminator to assign importance weights to the latent points, thus defining a new distribution in the latent space. Subsequently, rejection sampling on the newly defined latent distribution is applied to obtain inputs for a generator network. By treating the problem in the latent space, the paper introduces latentRS method that compares favourably to several existing methods in terms of computational complexity for generating a sample. The authors propose one more method, latentGA, following the path in the latent space that maximizes the learned importance weights. The paper also discusses the limitations of the previously proposed methods and presents their empirical comparison on several datasets and metrics.

The method is concise and straightforward to be applicable by a broad community of ML practitioners. One of the proposed methods, latentRS, offers a significant speedup at the inference stage compared to analyzed methods while being similar in performance metrics. The paper also raises an interesting question of whether the existing enhanced sampling methods help when the target distribution is not sufficiently disconnected.

However, there are several weaknesses in the experiments which lead to questioning the claims. While the paper's claim of careful comparison with the existing methods and the discussion of existing methods' limitations is indeed well-presented, the paper neglects the already recurring standard experiments for such methods or brushes them under the Appendix section. The generated samples from the Appendix figure for the mixture of Gaussians with n>9 show that the results are not as promising as the same experiments in the literature (the 'fake' clouds are not as nicely located on top of the true ones). n=25 is a recurring setting and seems to be a standard check for algorithms that refine GAN's sampling (e.g. DRS, DOT, DDLS). Table 2 in the Appendix lacks computer metrics for existing methods since it would demonstrate the tangible interpretable difference in this setting between comparable methods.

The paper misses some essential experiments to be faithfully compared with existing methods. It would be helpful to see the Swiss Roll experiment and the statistics on recovered modes and quality on 25 Gaussians for all the considered methods. As for more realistic image spaces, the CIFAR10 is a dataset that represents an undoubtfully disconnected manifold, and it has more potential to show the advantage of the proposed methods.

Returning to the presented empirical study, these too raise a number of questions. The IPR results in Table 1 (and Table 3 in the Appendix) do not show consistent advantages for the proposed methods over the existing ones. It either favours latentRS or latentGA in terms of precision or recall alone, not both at the same time. It's understandable that when maximizing importance weights with latentGA we get higher precision; we force the generated samples to stay within true points at the cost of their diversity (which can also be seen in the synthetic experiment with Gaussians), so I guess the method is highly reliant on the hyperparameter m, which controls the 'conservativeness' of the trained importance weights network. It would be helpful to see an ablation study for the hyperparameters.

Given all the above, I am leaning towards a reject and my main concerns are as follows. The experiment with 25 Gaussians doesn't show as much improvement in sampling as existing methods implying there might be little effect in real-world datasets. I believe that the proposed methods have not been faithfully compared to the existing methods. There is no ablation study on the hyperparameters of the proposed methods.

While the argument about one class CelebA has grounds, the DRS technique shows that it improves face generation by producing less warped nightmare-like faces. Thus, better GAN sampling techniques should ideally not only help avoid empty regions in the latent space between the nodes (inject disconnectedness) but also grasp the shape of those modes. In this regard, using an energy-based model for the latent variable might be an apt direction [5].

The authors state that they use image embeddings from corresponding classification networks for each of their datasets, but how do they obtain image embeddings for CelebA?

Some minor points — the notation for the proposal and true distribution we want to sample from in section 3 is a bit confusing as the hat is usually used to denote an approximation. Also, the submission has quite a few typos — it needs proofreading.

References:
[1] Azadi, Samaneh, et al. "Discriminator rejection sampling." arXiv preprint arXiv:1810.06758 (2018).

[2] Turner, Ryan, et al. "Metropolis-hastings generative adversarial networks." International Conference on Machine Learning. 2019.

[3] Neklyudov, Kirill, Evgenii Egorov, and Dmitry P. Vetrov. "The Implicit Metropolis-Hastings Algorithm." Advances in Neural Information Processing Systems. 2019.

[4] Tanaka, Akinori. "Discriminator optimal transport." Advances in Neural Information Processing Systems. 2019.

[5] Che, Tong, et al. "Your GAN is Secretly an Energy-based Model and You Should use Discriminator Driven Latent Sampling." arXiv preprint arXiv:2003.06060 (2020).

---

> ### Author Response · Authors · 2020-11-16
> **Authors response to AnonReviewer1**
>
> First, we would like to thank you for reviewing our paper and for your valuable comments. We will try to answer your main concerns.
>
> 1) "The paper misses some essential experiments to be faithfully compared with existing methods. It would be helpful to see the Swiss Roll experiment and the statistics on recovered modes and quality on 25 Gaussians for all the considered methods."
> The main advantage of our method arises in high-dimensions since it relies on minimizing the Wasserstein distance, which is robust and well-behaved even when the supports of distributions do not intersect. On the other side, our concurrent methods rely on density-ratio estimation which have an inherent weakness in high-dimension. Indeed, in a  high-dimensional space $X$, the intersection of target and generated manifolds has a measure of 0 in $X$ (see [1]). Thus, density ratios are trivial: always 0 on the support of the generated distribution. This requires to strongly regularize the classifier that estimates the density ratios, but then the density ratios are not accurate anymore.
> For clear comparisons with other works, we have updated the paper with according experiments on Swiss Roll and Mixture of 25 Gaussians (Figure 6 and Table 3 in Appendix). It supports our point: latentRS surpasses or matches the other methods on EMD, which is an empirical measure of the Wasserstein distance between generated and target distributions. Methods based on density-ratios do not always improve the goodness of fit measured in terms of EMD.
> [1] Arjovsky, Martin, and Leon Bottou. "Towards Principled Methods for Training Generative Adversarial Networks." ICLR 2017.
>
> 2) "While the argument about one class CelebA has grounds, the DRS technique shows that it improves face generation by producing less warped nightmare-like faces."
> We performed this experiment quite fairly and were not able to observe significant improvements over the GAN baseline. As mentionned in the Appendix, for DRS/SIR/MH-GAN, we selected the best from two different discriminators: 1) the 1-Lipschitz discriminator from the adversarial training, which is then fine-tuned with binary cross-entropy; 2) a non-regularized discriminator trained from scratch with binary cross-entropy. We selected the best from several checkpoints and reported the results. There is no significant improvement in terms of Precision, Recall, EMD or Avg. Hausd.
> However, we see that our method LatentRS slightly improves EMD and Avg. Hausd.
>
> 3) "The authors state that they use image embeddings from corresponding classification networks for each of their datasets, but how do they obtain image embeddings for CelebA?"
> For CelebA, we use the standard Frechet Inception Distance with Inception Network embeddings, and VGG16 pre-trained from PyTorch for the other metrics: Precision, Recall, Avg. Hausdorff and EMD.

---

### Official Review · AnonReviewer2 · 2020-10-29
**This work proposes a new rejection sampling technique for improving the quality of images generated from GANs**

**Rating:** 4
**Confidence:** 3

**Review:**

## Summary:
The paper proposes a new algorithm for improved sampling of GANs. Since GANs are continuous functions that act on a connected latent space, they will have trouble learning distributions whose support is disconnected (for e.g., clustered data). The proposed method tries to fix this issue and is motivated by rejected sampling. However, instead of using density based algorithms for rejecting samples, the authors take a fixed pre-trained generative model and train a neural network that learns to reject samples from the latent space.

## Significance:
The problem is well motivated and seems significant. Learning distributions with disjoint support can be difficult using the traditional GAN training, and this paper addresses this problem by learning which areas in the latent space must be avoided.

## Quality:
While the problem is significant, I find that the proposed method has some weaknesses in its formulation and empirical evaluation. Please see the section "Cons" below.

## Originality:
The proposed method seems sufficiently novel, but I am not familiar enough with this area to know if something closely related has been done before.

## Pros:
1. On synthetic data, the proposed method can capture modes better than the considered baselines.
1. The proposed technique produces better quality samples on GANs trained on CelebA.

## Cons:
1. While some of the experiments are convincing, I do not buy some of the arguments made in the paper. Specifically, under eqn (3), it is argued that if the GAN $G$ is kept fixed and the following adversarial training is performed for a classifier $w_\phi$ and discriminator $D_\alpha$, then the following procedure:
$$ \sup_{\alpha \in A} \inf_{\phi \in \Phi} E_{x \sim \mu} D_\alpha(x) - E_{z \sim Z} [ w_\phi(z) \cdot D_\alpha( G(z) )]$$
will produce a new distribution on $z$ which is $\widehat{\gamma}(z) \propto \gamma(z) w_\phi(z)$.
There is no proof of this claim, and I further suspect that this claim in not correct.

1. There exist several baseline methods that consider the problem of mode collapse. Examples like PAC-GAN [Lin et al 2017] have shown to be effective, and are also provably good. Other examples include those considered in table 5 of https://arxiv.org/pdf/2010.00654v1.pdf
For the Stacked-MNIST dataset, the algorithms listed in table 5 seem to have much better mode coverage than the algorithms in Table 1 of this submission.  While the paper I have linked is very recent, the baseline algorithms considered in the paper are published works from 2017 onwards.
Comparing to these algorithms would make this paper much stronger.

1. There are a lot of unsubstantiated claims. Some include:
    1. Modifying the loss in equation 3 gives a new distribution $\widehat{\gamma}$ defined underneath equation 3.
    1. At the bottom of page 5, the authors remark "In our method, on the contrary, we are still looking for the discriminator maximizing the Integral Probability Metric (Müller, 1997) in equation 3, linked to optimal transport." .
In equation 3, the authors take the dual optimization problem for estimating the Wasserstein distance, and modify it. With this modification, optimizing equation 3 will not give the Wasserstein distance any more.

## Minor:
1. Should there be a quanitifer in eqn (3) restricting $D_\alpha$ to $1-$Lipschitz models?

---

> ### Author Response · Authors · 2020-11-16
> **Authors response to AnonReviewer2**
>
> First, we would like to thank you for reviewing our paper and for your valuable comments. We will try to answer your two main concerns.
>
>  1) " It is argued that the procedure will produce a new distribution $\widehat{\gamma}(z) \propto \gamma(z) w_{\varphi}(z)$. There is no proof of this claim, and I suspect that this claim is not correct. "
> This is not a claim and is the definition of our method: it learns an importance weighting function $w_\varphi$, that assigns an importance weight $w_\varphi(z) \in R$ to each point $z \sim \gamma(z)$ in the latent space. Given that the function is normalized, i.e. $\int_z \gamma(z) w(z) dz = E_\gamma w_\varphi=1$, then $\widehat{\gamma}(z) = \gamma(z) w_\varphi(z)$ is a well-defined probability distribution.
> In practice, we add a strong regularization term forcing $\widehat{\gamma}$ to define a normalized distribution (self-normalization). We track the quantity $(\{E}_{z\sim Z} [w_\varphi (z)] -1)^2$ during training, and it is indeed 0 after a few training steps, which gives us a guarantee that $\widehat{\gamma}$ defines a distribution.
>
> 2) "Optimizing equation 3 will not give the Wasserstein distance any more."
> Actually, the method is indeed estimating and minimizing the Wasserstein distance. The Wasserstein distance consists in estimating two integrals. On the left side (target distribution), we do not bring any modification to the standard formulation. On the right side (generated distribution), the generated distribution is modified by the learnt importance weights, which is a classical way of performing importance sampling.
>
> 3) "Minor: Should there be a quanitifer in eqn (3) restricting  to 1-Lipschitz models?"
> In Related Work, equation 1, we mention that $D_\alpha \in A$ are functions restricted to the class of 1-Lipschitz functions.
>
> Furthermore, as asked by AnonReviewer1, we have updated the paper with according experiments on Swiss Roll and Mixture of 25 Gaussians (Figure 6 and Table 3 in Appendix). It supports our point: latentRS surpasses or matches the other methods on EMD, which is an empirical measure of the Wasserstein distance between generated and target distributions. Methods based on density-ratios do not always improve the goodness of fit measured in terms of EMD.

---

> > ### Comment · AnonReviewer2 · 2020-11-16
> > **Still don't understand**
> >
> > 1. But $w_\phi$ multiplies the test function $D$, so it doesn't modify the distribution of $z$ the way you want it to. I think you want to say that in the second integral, you draw from a distribution $Z'$, which is a weighted version of $Z$. That's *completely* different from what you have in  equation (3).
> >
> > 2. Can you give me an example of a function $w_\phi$ which satisfies $E_{z} [ (w_\phi(z) - 1)^2 ] = 0$ ? The only function that satisfies this is $w_\phi(z) = 1 $ almost surely over $z$. In your experiments you draw a finite number of $z$ from a continuous probability distribution, which is a measure zero set, and does not contradict my claim that $w_\phi(z)=1$ almost surely.

---

> > > ### Author Response · Authors · 2020-11-16
> > > **Authors response**
> > >
> > > 1. Yes, it multiplies by a factor which corresponds to the importance weight. Let us imagine that we want to estimate the quantity $\int \widehat{\gamma}(z) f(z) dz$, but we only can sample from $\gamma$. Then, we can rewrite the quantity of interest as follows: $\int \widehat{\gamma}(z) f(z) dz = \int \gamma(z) \frac{\widehat{\gamma}(z)}{\gamma(z)} f(z) dz = \int \gamma(z) w_\varphi(z) f(z) dz \approx \frac{1}{n} \sum_{i=1,...,n} w_\varphi(z_i) f(z_i)$; where each $z_i$ is drawn from $\gamma$. It means that we sample from $\gamma$ and we re-weight each sample by its importance weight. It corresponds to what we propose in equation (3). That is the standard use of importance sampling.
> > >
> > > 2. Yes, you can check Figure 1.d for example. This is a visualization of $w_\varphi(z)$ after training on a mixtures of 4 gaussians. The function is defined on the square [-1,1]x[-1,1]; its minimum is 0 and maximum is 2.

---

> > > > ### Author Response · Authors · 2020-11-17
> > > > **Authors response**
> > > >
> > > > 2. There was a typo in our first answer. As written in the paper, the quantity that we track is $(\{E}_{z\sim Z} [w_\varphi (z)] -1)^2$.  This quantity goes nearby 0 after a few training steps, ensuring that the penalty term forces $\{E}_{z\sim Z} [w_\varphi (z)]  = 1$, so that $\widehat{\gamma}$ defines a normalized probability distribution.

---

> > > > > ### Comment · AnonReviewer2 · 2020-11-23
> > > > > **Multiple concerns remain**
> > > > >
> > > > > I share a lot of concerns with Reviewer 4. The justifications used for the learned function $w_\phi$ are too flimsy. The authors have a heuristic for ensuring that $w_\phi(z) \gamma(z)$ is a valid probability distribution. However, for a finite number of samples, their heuristic will have some error which the authors do not account for.
> > > > >
> > > > > This theoretical limitation could be overlooked, if the experiments were convincing enough. As I mentioned in my original review, there are explicit methods that can recover modes of a distribution, and the authors do not compare to any of them.
> > > > >
> > > > > Given the theoretical and experimental limitations of this work, I do not have a convincing enough reason to change my score.
> > > > >
> > > > > Additionally:
> > > > > I see you edited your original author rebuttal to fix the typo, which is fine. However, please add a note at the bottom to mark what the edits were. It's impossible to keep track of changes, and it invalidates later discussions.

---

> > > > > > ### Author Response · Authors · 2020-11-24
> > > > > > **Authors Response**
> > > > > >
> > > > > > We share the answer to Reviewer 4's concerns.
> > > > > >
> > > > > > A neural network can be considered as a bounded Lipschitz function (e.g. a ReLU network is piece-wise linear). Thus, it comes from [1, Theorem 3.1] that the empirical measure $\frac{1}{n} \sum w_\varphi(z)$ converges to $E_{z\sim Z}[w_\varphi(z)]$. In our setting the consistency is an immediate consequence of theorems on Donsker's classes for the Lipschitz cases.
> > > > > >
> > > > > > [1] van der Vaart, Aad W., and Jon A. Wellner Wellner. "Empirical processes indexed by estimated functions." Asymptotics: particles, processes and inverse problems. Institute of Mathematical Statistics, 2007. 234-252.

---

### Decision · Program_Chairs · 2021-01-07
**Final Decision**

**Decision:**

Reject

**Comment:**

The paper proposes to train a rejection sampler in the latent space of a GAN to learn disconnected data manifolds. Reviewers raised concerns about some theoretical aspects of the method as well as about the lack of larger scale datasets (ImageNet) in the experiments. Authors responded to these concerns but some of them still remain (including $\hat{\gamma}(z)$ not guaranteed to be a probability distribution and lack of more convincing experiments). I still think the work is promising, and encourage the authors to revise and resubmit the paper addressing these points highlighted by the reviewers.